# Nutritional Factors Modulating Alu Methylation in an Italian Sample from The Mark-Age Study Including Offspring of Healthy Nonagenarians

**DOI:** 10.3390/nu11122986

**Published:** 2019-12-06

**Authors:** Robertina Giacconi, Marco Malavolta, Alexander Bürkle, María Moreno-Villanueva, Claudio Franceschi, Miriam Capri, P. Eline Slagboom, Eugène H. J. M. Jansen, Martijn E. T. Dollé, Tilman Grune, Daniela Weber, Antti Hervonen, Wolfgang Stuetz, Nicolle Breusing, Fabio Ciccarone, Michele Zampieri, Valentina Aversano, Paola Caiafa, Laura Formentini, Francesco Piacenza, Elisa Pierpaoli, Andrea Basso, Mauro Provinciali, Maurizio Cardelli

**Affiliations:** 1Advanced Technology Center for Aging Research, IRCCS INRCA, 60121 Ancona, Italy; r.giacconi@inrca.it (R.G.); m.malavolta@inrca.it (M.M.); lauraformentini26@gmail.com (L.F.); f.piacenza@inrca.it (F.P.); e.pierpaoli@inrca.it (E.P.); a.basso@inrca.it (A.B.); m.cardelli@inrca.it (M.C.); 2Molecular Toxicology Group, Department of Biology, Box 628, University of Konstanz, 78457 Konstanz, Germany; alexander.buerkle@uni-konstanz.de (A.B.); maria.moreno-villanueva@uni-konstanz.de (M.M.-V.); 3Human Performance Research Centre, Department of Sport Science, Box 30, University of Konstanz, 78457 Konstanz, Germany; 4Lobachevsky State University of Nizhny Novgorod, 603950 Nizhny Novgorod, Russia; claudio.franceschi@unibo.it; 5DIMES-Department of Experimental, Diagnostic and Specialty Medicine, Via S. Giacomo, 12, ALMA MATER STUDIORUM, University of Bologna, 40126 Bologna, Italy; miriam.capri@unibo.it; 6Section of Molecular Epidemiology, Leiden University Medical Centre, 2333 ZA Leiden, The Netherlands; p.slagboom@lumc.nl; 7Centre for Health Protection, National Institute for Public Health and the Environment, PO Box 1, 3720 BA Bilthoven, The Netherlands; eugenejansen99@gmail.com (E.H.J.M.J.); martijn.dolle@rivm.nl (M.E.T.D.); 8Department of Molecular Toxicology, German Institute of Human Nutrition Potsdam-Rehbruecke (DIfE), 14558 Nuthetal, Germany; scientific.director@dife.de (T.G.);; 9NutriAct-Competence Cluster Nutrition Research Berlin-Potsdam, 14458 Nuthetal, Germany; 10Medical School, University of Tampere, 33014 Tampere, Finland; antti.Hervonen@uta.fi; 11Institute of Nutrition, University of Hohenheim, 70599 Stuttgart, Germanynicolle_breusing@yahoo.de (N.B.); 12IRCCS San Raffaele Pisana, Department of Human Sciences and Promotion of the Quality of Life, San Raffaele Roma Open University, 00166 Rome, Italy; ciccarone@bio.uniroma2.it; 13Faculty of Pharmacy and Medicine, Department of Experimental Medicine, Sapienza University of Rome, 00161 Rome, Italy; michele.zampieri@uniroma1.it (M.Z.); aversano@bce.uniroma1.it (V.A.); caiafa@bce.uniroma1.it (P.C.); 14Pasteur Institute-Fondazione Cenci Bolognetti, 00161 Rome, Italy

**Keywords:** Alu methylation, aging, longevity, nutrients, antioxidant

## Abstract

Alu hypomethylation promotes genomic instability and is associated with aging and age-related diseases. Dietary factors affect global DNA methylation, leading to changes in genomic stability and gene expression with an impact on longevity and the risk of disease. This preliminary study aims to investigate the relationship between nutritional factors, such as circulating trace elements, lipids and antioxidants, and Alu methylation in elderly subjects and offspring of healthy nonagenarians. Alu DNA methylation was analyzed in sixty RASIG (randomly recruited age-stratified individuals from the general population) and thirty-two GO (GeHA offspring) enrolled in Italy in the framework of the MARK-AGE project. Factor analysis revealed a different clustering between Alu CpG1 and the other CpG sites. RASIG over 65 years showed lower Alu CpG1 methylation than those of GO subjects in the same age class. Moreover, Alu CpG1 methylation was associated with fruit and whole-grain bread consumption, LDL2-Cholesterol and plasma copper. The preserved Alu methylation status in GO, suggests Alu epigenetic changes as a potential marker of aging. Our preliminary investigation shows that Alu methylation may be affected by food rich in fibers and antioxidants, or circulating LDL subfractions and plasma copper.

## 1. Introduction

DNA methylation represents an epigenetic mechanism involved in the regulation of several biological processes, including gene expression, genomic stability and parental imprinting [1]. Most of DNA methylation in human genomic DNA occurs in the repetitive sequences, among which the Alu elements (short-interspersed nuclear elements with a length of about 300 bp) represent approximately 11% of the genome and contains 23% of its methylation sites [2,3,4]. Based on their evolutionary history, Alu elements are divided in subfamilies of different evolutionary age, classified in the major groups Alu J, Alu S and Alu Y [3,4]. The measurement of methylation across repetitive DNA elements has been shown to be associated with global methylation levels [5]. Alu elements are transcribed at low levels by RNA polymerase III into Alu RNA copies, that contribute to retrotransposition, a mutational mechanism active in the germline as well as in somatic cells [6,7,8].

In normal tissues most Alu elements are methylated and transcriptionally inactive [9,10]. This condition has been proven to stabilize the genome by preventing accumulation of DNA damage [11], whereas hypomethylation of Alu elements is associated with genomic instability [12].

Alu hypomethylation has been found during aging [13] and it has been associated with the severity of age-related diseases such as diabetes [14], cancer [15], osteoporosis [16] and cardiovascular diseases [17]. Interestingly, hypomethylation and epigenetic age are delayed in centenarians’ offspring [18,19], which are considered an optimal model to study biomarkers associated with human longevity. Indeed, centenarians’ offspring show decreased epigenetic age of PBMCs than those of age matched controls [19,20] and display a greater genomic stability and lower DNA damage, that may contribute to explain their delayed onset of age-related diseases and prolonged survival [18]. In addition to a favorable genetic background that might delay hypomethylation, other studies show that other environmental and lifestyle factors such as smoking, alcohol consumption, physical activity, and diet [21,22], may influence Alu methylation in blood DNA. Folate, vitamin B2, vitamin B6, vitamin B12, choline, and methionine are nutrients involved in the 1-carbon cycle for the production of S-adenosyl-methionine (SAM), the universal methyl donor, required for DNA methylation [23,24]. Therefore, a nutritional deficiency or excess of these nutrients may induce aberrant DNA methylation patterns, adversely affecting gene expression and susceptibility to diseases. Recently, other nutrients such as circulating carotenoids, zinc, selenium and vitamin E have been shown to impact global DNA methylation. [25,26,27,28]. This phenomenon may in part be attributed to their influence on oxidative stress which, in turn, may alter the activity of DNA methyltransferases (DNMTs) and promote DNA hypomethylation [28,29] 

However, there are still very few studies evaluating the effect of antioxidant nutrients on Alu DNA methylation in the context of aging and longevity. Therefore, the aim of the present study is to perform a preliminary investigation on the relationship between circulating metal trace elements, lipids, antioxidants such as carotenoids and several nutritional factors and Alu methylation in the general population and in offspring of healthy nonagenarians.

## 2. Materials and Methods

### 2.1. Study Population and Blood sample Collection

DNA samples used in the present work were randomly selected from an Italian cohort of randomly recruited age-stratified individuals from the general population (RASIG; n = 60; mean age = 57 ± 12 years) and offspring of healthy nonagenarians (originally recruited in the GeHA Study) (GO [GeHA offspring]; n = 32; mean age = 65 ± 5 years) that have been recruited in the MARK-AGE project [30,31,32]. Details of the recruitment procedures and of the collection of anthropometric, clinical and demographic data, as well as of laboratory parameters assays have previously been reported [30,31].

Anticoagulated whole blood, obtained by phlebotomy after overnight fasting, was collected. Prepared samples of plasma, serum and whole blood from Italian recruitment center were shipped to the MARK-AGE Biobank located at the University of Hohenheim, Stuttgart, Germany. From the Biobank, coded samples were subsequently sent to the Scientific and Technological Pole of INRCA of Ancona, Italy, on dry ice, where they were stored at −80 °C until use. Local Ethics Committees approval for the recruitment of Italian participants in MARK-AGE project was obtained by S.Orsola-Malipighi Hospital Committee (approved the project on 24/06/2008 code n. 75/2008/U/Tess) and AUSL—Independent Ethical Commettee (Emilia-Romagna Region) approved on 01/04/2009—Prot num 346/CE; CE: 09007.

### 2.2. NMR Analysis of Lipoprotein Subclasses

The NMR lipoprotein subclass profile was determined using a 400-MHz proton NMR analyzer at LipoScience (http://www.liposcience.com). In brief, the NMR method uses the characteristic signals broadcast by lipoprotein subclasses of different size as the basis of their quantification [33]. Each measurement produces the signal amplitudes of lipoprotein subclasses that allows one to estimate the total LDL and HDL particle concentration as well as their subclasses including large (21.3–23.0 nm) and small (18.0–21.2 nm) LDL particles. For this estimation, conversion factors are used to relate these signal amplitudes to particle concentration or lipid mass concentration units, which were obtained from NMR and chemical lipid analyses of a set of purified subclass standards. Particle concentrations (nmol/L) were derived for each subclass standard by measuring the total concentration of core lipid (cholesterol ester plus triglyceride) and dividing the volume occupied by these lipids by the core volume per particle calculated from knowledge of the particle’s diameter. Triglyceride concentrations were calculated from NMR subclass measurements. Average particle sizes (diameter in nanometres) of LDL and HDL were determined by weighting the relative mass percentage of each subclass according to its diameter. Note that due to the relatively small range in diameters for various LDL and HDL lipoprotein subclasses and the weighting, limited absolute inter-individual differences exist in mean particle sizes (resulting in small standard deviations and standard errors). 

### 2.3. Metal Trace Element Determination in Plasma Samples

The plasma concentrations of Selenium (Se), zinc (Zn), iron (Fe) and copper (Cu), were determined by a Thermo XII Series ICP-MS (Thermo Electron Corporation, Waltham, MA, USA) by adapting methods used for the measurement of trace elements in human plasma with slight modifications [34,35] In detail, plasma samples were diluted 1:10, with a diluent containing 0.1% triton and 0.15% HNO3, to ensure that trace elements are maintained in solution and to favour the washout of these elements between samples. External calibration solutions containing Se, Fe, Zn, and Cu (blank to 2000 ppb) were prepared by serial dilution of a parent multi-element solution (1000 ppm for Zn and Cu) (VHG Labs, Manchester, NH 03103, USA), using the same diluent used for the samples. Rhodium (Rh), at 200 ng/mL was used as internal standard. Data were acquired for ^80^Se, ^56^Fe, ^66^Zn, ^65^Cu. 

Quality of the analysis was assured by assessment of “quality standard samples” (SERONORM™ TRACE ELEMENT SERUM, Sero AS, Billingstad, Norway). Se, Fe, Zn and Cu levels of the quality standard samples were within 10% of the certified levels, as previously reported [35]. The instrument was operated with a Peltier cooled impact bead spray chamber, single piece quartz torch (1.5 mm i.d. injector) together with Xi interface cones and a Cetac-ASX 100 autosampler (CETAC Technologies, Omaha, NE, USA). A Burgener Trace nebulizer was used as this device does not block during aspiration of clinical samples. The instrument was operated in standard mode (non-CCT), using 1400 WRF power, 1.10 L min^−1^ nebulizer gas flow, 0.70 L min^−1^ auxiliary gas flow, 13.0 L min^−1^ cool gas flow, 70 ms dwell time, 30 s sample uptake and 35 s wash time (2 repeats per sample).

### 2.4. Systemic Inflammation Parameters

C-reactive protein (CRP), homocysteine, fibrinogen and adiponectin were measured as markers of inflammatory status [36]. In particular, the assays for high-sensitive CRP (HS-CRP), fibrinogen and adiponectin were turbidimetric assays (immunoprecipitation), whereas homocysteine was an enzymatic cycling assay. The HS-CRP assay was obtained from Beckman Coulter (Woerden, The Netherlands), the fibrinogen and homocysteine assays were obtained from Dialab (Neudorf, Austria). The measurements of HS-CRP, homocysteine and fibrinogen were performed on a LX20 autoanalyzer (Beckman Coulter). Adiponectin levels were assessed by time resolved fluorescent immunoassay utilizing R&D systems monoclonal antibodies (Mab 10651 & BAM1065) on an AutoDELFIA^®^ automated immunoassay system (PerkinElmer, Turku, Finland).

### 2.5. Determination of Total Glutathione and Total Free Cysteine in Whole Blood

Total glutathione and total free cysteine in whole blood were measured by RP-HPLC with UV-Vis detection after reduction and modifications as previously reported [37].

### 2.6. Determination of Ascorbic Acid and Uric Acid in Plasma

Plasma ascorbic acid and uric acid were analyzed by RP-HPLC and UV detection after reduction with tris-(2-carboxyethyl)-phosphine. The details of the method have been previously reported [37]. 

### 2.7. Determination of Total Carotenoid Plasma Levels

The carotenoids (α-carotene, β-carotene, β-cryptoxanthin, lutein, zeaxanthine, and lycopene, α-tocopherol, γ-tocopherol and retinol) were analyzed in plasma and determined simultaneously by RP-HPLC with UV and fluorescence detection as previously described [38]. 

### 2.8. DNA Extraction and Bisulfite Treatment

Genomic DNA was extracted from whole blood using the QIAamp DNA Blood Mini Kit (Qiagen, Hilden, Germany). Samples were processed for DNA isolation following the manufacturer’s instructions. Concentration and purity of isolated DNA samples was measured using a Nanodrop spectrophotometer. Sodium bisulfite conversion of DNA (50 ng) was performed by the EpiTect Bisulfite Kit (Qiagen), purified with the same kit, eluted in 20 uL of EB buffer and stored at −20 °C. 

### 2.9. Bisulfite Pyrosequencing Analysis of Alu DNA Methylation

Alu methylation analysis was quantitatively performed on the bisulfite-treated DNA using pyrosequencing. The polymerase chain reaction (PCR) primers and assay conditions were the same described by other authors [39], with slight modifications. In brief, about 5 ng of bisulfite-converted DNA samples were amplified in 20 uL reaction volume with 1 pmol of biotin-labeled forward primer (biotin-5’-TTTTTATTAAAAATATAAAAATT-3’), 1 pmol of reverse primer (5’-CCCAAACTAAAATACAATAA-3’), (1×) PyroMark PCR Master Mix, 2 uL of Q-solution (Qiagen), 2 uL of Coral Load (Qiagen). PCR was conducted with the following steps: predenaturation 15 min at 95 °C; 5 cycles with 30 s at 94 °C, 30 s at 43.5 °C and 30 s at 72 °C; 45 cycles with 30 s at 94 °C, 30 s at 40 °C and 30 s at 72 °C; final extension at 72 °C for 10 min. Confirmation of the quality of the PCR products and their freedom from contamination was established on 2% agarose gels with gel red staining. After purification of the biotinylated strand using Sepharose beads on a PyroMark Vacuum Prep Workstation (Qiagen), pyrosequencing was carried out by using PyroMark Q24 reagents and the PyroMark Q24 System (Qiagen) according to manufacturer’s instructions. The sequencing primer was 5’-AATAACTAAAATTACAAAC-3’. The percentage of methylation (% 5-mC) was expressed for each CpG site as 5-mC divided by the sum of methylated and unmethylated cytosines. The technical reproducibility was tested by making triplicate PCR reactions and pyrosequencing assays for 20 samples, and resulted in mean standard deviation among replicates of a same sample <2%. At least 2 wells for each pyrosequencing plate were used for control pyrosequencing reactions on a control oligonucleotide (PyroMark Control Oligo, Qiagen), aimed to verify the correct functioning of the pyrosequencing system and reagents. Methylated and unmethylated genomic DNA samples (Epitect Control DNA Set, Qiagen) were used as additional controls to check reaction optimization, and to construct calibration curves obtained by mixing different percentages of methylated and unmethylated control DNA sample. Further information concerning the pyrosequencing method, including a schematic representation of an Alu element (Appendix A), the position of the assayed CpGs (Appendix A), the exact sequences to analyze and nucleotide dispensation order (Appendix A), and the results of the calibration curves (Appendix A), can be found in Appendix A.

Five CpG positions were included in the pyrosequencing assay and considered for the methylation analysis. In order to verify the exact sequence and composition of the Alu sequences recognized as targets by the primers used in this assay for PCR and for pyrosequencing, the complementarity of the primers with the (bisulfite-converted) consensus sequences of Alu subfamilies was checked (see Appendix A). The following subfamilies of the “Alu S” group resulted to be perfectly recognized (based on their consensus sequence) by all the primers used in the assay: Alu Sx, Alu Sg, Alu Sz, Alu Sz6, Alu Sg1, Alu Sg4, Alu Sq, Alu Sq4, Alu Sx1, Alu Sg7, Alu Sq2, Alu Sq10 and Alu Sp. Not all the target Alu subfamilies share all the five CpG sites included in the assay, because some of the CpG sites are mutated in part of the subfamilies (see Appendix A). On the whole, the target Alu subfamilies include over half a million elements, located mainly in gene introns (57.1%) (Appendix A). Near five percent of the elements overlaps with transcription factor binding sites in intergenic regions, while a smaller fraction (2.1%) overlaps with gene exons, while. Albeit members of Alu S subfamilies have a percentage of identity with their consensus sequence which exceed 90% when excluding CpG sites [40], sequence variation of single Alu elements with respect to their subfamily’s consensus is expected to reduce the number of elements which are actually targeted by the assay. Albeit the exact number of Alu elements actually contributing to the methylation values has not been estimated in the present work, Yang and coauthors (using bisulfite PCR primers partially overlapping with those used in the present assay) estimated this number in the order of 15,000 [41].

### 2.10. Statistical Analysis

Subject characteristics were reported as mean ± standard error of the mean (SEM) or percentages for continuous and categorical variables, respectively. For continuous variables, normal distribution was verified by the 1-sample Kolmogorov–Smirnov test. All the variables not normally distributed were log-transformed. Exploratory factor analysis was used to evaluate correlations and potential clustering among Alu CpGs. Differences among groups were checked by One-way Analysis of Variance or Kruskal Wallis test for continuous variables and Pearson’s χ^2^ test for categorical variables. A Generalized Linear model was used to evaluate differences in Alu methylation in relation to age classes between GO and RASIG. Automatic regression analysis using forward stepwise method was carried out to explore the main predictors of Alu methylation. The most important predictors were included in a generalized linear model that included the effect of gender, subject group (GO or RASIG) and age to confirm the associations. All the analyses were performed using the SPSS/Win program (version 22.0; SPSS Inc., Chicago, IL, USA).

## 3. Results

### 3.1. Characteristics of Subjects

Table 1 summarizes the main characteristics of RASIG and GO subjects used in this study and subdivided in age classes. No differences were found in gender, smoking habit, alcohol consumption and in the laboratory parameters between younger (55–64 years) and older (65–75 years) GO subjects. In contrast older RASIG (65–75 years) had lower platelet levels than subjects in the 55–64 and 45–54 age classes. Moreover, RASIG in the age range 55–64 years showed higher total and LDL cholesterol than younger RASIG (under 54 years). Comparisons among RASIG and GO showed differences only in total and LDL cholesterol in the 55–64 age class. 

### 3.2. Alu Methylation

Factor analysis revealed a strong (*p* < 0.001; *r* > 0.3) and positive correlation among Alu CpG 2, 3, 4 and 5, whereas Alu CpG 1 displayed only weak correlations (*p* < 0.05; *r* < 0.35) with CpG 2, 4 and CpG 5 (Appendix A). Indeed, principal component analysis extracted two components with eigenvalues greater than 1.0 (Appendix A). Component 1 is mainly characterized by positive factor loadings of CpGs 2, 3, 4 and 5 whereas component 2 is mainly characterized by CpG1 (Appendix A). This exploratory analysis suggests investigating Alu CpG 1 separately from the other Alu CpGs. In RASIG subjects the percentage of Alu methylation at CpG 1 showed a decline in the 65–75 age class compared with 55–64 age class (Figure 1). Conversely, the mean methylation levels of the others Alu CpGs were increased in the 55–64 age class compared to the 35–44 age class (Appendix A).

Interestingly, GO subjects in the older age class showed a Alu CpG1 methylation similar to younger GO, but with a higher Alu CpG1 methylation level as compared to RASIG over 65 years old (Figure 1; *p* < 0.05). These results suggest that the age-related Alu CpG1 methylation decline is delayed in GO and this is consistent with the assumption that GO have a lower “biological” age than RASIG [19,20], As a consequence, Alu CpG1 methylation might be a potential biomarker of healthy aging for persons over 65 years.

### 3.3. The Association Between Alu CpG1 Methylation Levels and Nutritional, Metabolic and Inflammatory Factors

An automatic regression analysis was run in the whole sample including Alu CpG 1 methylation levels as dependent variable and the following factors as independent variables: age, gender, subject group (GO and RASIG), plasma metal trace elements (copper, selenium, zinc and iron), serum lipid markers (VLDL1Cholesterol, VLDL1Triglycerides, VLDL2Cholesterol, VLDL2Triglycerides, LDL1Cholesterol, LDL1 Triglycerides, LDL2Cholesterol, LDL2Triglycerides, HDL1Cholesterol, HDL1Triglycerides, HDL2 Cholesterol, HDL2Triglycerides), systemic inflammation parameters (CRP, homocysteine, uric acid, fibrinogen and adiponectin), insulin, glucose, free fatty acids, albumin, antioxidant nutrients (ascorbic acid, α-tocopherol, γ-tocopherol, cysteine, lutein, lycopene, α-carotene, β-carotene, β-cryptoxanthin, zeaxanthin, retinol, total glutathione), consumption of fruit, fish, whole-grain bread, white bread, brown bread, dairy products, vitamins, meat, eggs, fried fries and vegetables. Variables that were found to be among the most important predictors (*p* < 0.05) of Alu CpG 1 methylation are reported in Table 2. 

A generalized linear regression model, including the main predictors of Alu CpG 1 methylation found here, confirmed a positive association with copper and LDL2 cholesterol levels (Table 3). Moreover, we found that people with the lower consumption of whole-grain bread (<1 serv./week) and fruit (<1 serv./day) display a lower level of Alu CpG 1 methylation (Figure 2).

Repeating the same analysis for the mean methylation levels of the others Alu CpGs (CpG 2–5), we detected that mean methylation levels were associated with higher consumption of meat (*p* < 0.01) and lower consumption of brown bread (*p* < 0.05) (Appendix A). Moreover, conversely to CpG1, increased mean methylation of CpG 2–5 was also associated with age (*p* < 0.05) and was not found to be dependent on subject group (GO, RASIG).

## 4. Discussion

Previous studies reported an age-dependent decline in genomic DNA methylation, both in global DNA and in Alu elements [13,14,17]. DNA Alu hypomethylation may promote an enhanced retrotransposon activity and genomic instability [12] and is associated with the severity of some age-related diseases [14,15,16,17]. In our study, we observed that Alu methylation at the CpG1 site do not display marked age-related changes, but lower levels of methylation at this site were found in elderly subjects over 65 years compared to the age class 55–64 years. Regarding the other CpGs, we observed only a significant increase in the 55–64 years age class compared to the youngest age class (35–44). This last finding would appear to be in contrast with previous results, where young subjects have higher Alu methylation levels than older ones [18]. However, the age range of young subjects was different (17–34 vs. >35 years in our study) and the differential methylation of different CpGs sites was not taken into account. 

In addition, differences in the geographical area, as well as the lifestyle and dietary habits of the subjects could explain these discrepancies. Other authors show a lack of age-associated Alu hypomethylation in cellular DNA from peripheral blood, while a decline of Alu methylation with age was found in circulating cell-free DNA [42]. A non-linear change with aging may in part explain the discrepancies observed by different groups. Moreover, the use of a single variable to represent the mean Alu methylation of different CpGs may result in a loss of information, if not all of the CpG sites are highly correlated. In the present work we have found a distinct clustering for Alu CpG1 which differs from the other four CpGs investigated. Most importantly, GO subjects do not display the reduction of Alu CpG1 methylation in the age-class 65.75 years compared to 55–64 years that was observed in RASIG. This finding could be the consequence of a delayed aging process in GO as observed in previous reports which support the theory of better preservation of several biological processes in centenarians’ offspring [18,20,43,44]. Hence, the preservation of Alu CpG1 methylation level in GO supports the idea that this methylation site may be a potential biomarker of healthy aging. The different results obtained for CpG1 and the other analyzed CpGs could have at least two possible explanations. The first one is a possible distinct functional effect of CpG1 methylation with respect to the methylation of the other CpG sites analyzed. The portion of Alu sequence analyzed in this assay overlaps with a response element for the p53 transcription factor, located around position 150 nearby the A’ box in the Alu sequence (the exact position of CpG1 is 155–156 in the Alu Sx consensus sequence, downloaded from the RepBase database, http://www.girinst.org/repbase/update/browse.php) [45]. Although at least part of the other CpGs are also located in the same p53 response element, it has been demonstrated that the effect of CpG methylation on the binding of the p53 transcription factor is highly variable depending on the exact position of the methylated CpG in the response element [46]. The second possible explanation of the specific results found for CpG1 compared to the other CpG sites is that the Alu elements possessing each of these CpG sites are not always the same elements. Indeed, an alignment of the target regions of the primers used for this bisulfite sequencing assay with the consensus sequences of the Alu subfamilies showed that the used primers are capable to target different subfamilies of the Alu S group. However, only part of these subfamilies shares all the five CpG sites analyzed in this study, while other subfamilies lack part of the CpGs (e.g., Alu Sg1 and Alu Sq subfamilies lack, respectively, CpG1 and CpG2). Other sequence differences may be present within each subfamily, considering that CpG sites are often mutated in Alu elements due to cytosine deamination [4]. Hence, it is possible that the results found for the CpG1 may reflect a specific functional role of Alu subfamilies or multiple Alu elements sharing this specific CpG site, but lacking the following CpG sites included in the assay.

In order to interpret the partial variation of Alu methylation observed in age classes, it may be useful to remark some aspects of the Alu genomic distribution. Albeit present in all the chromosomes and genomic regions, these repetitive elements have an uneven genomic density, being particularly abundant in G-C rich and gene-rich regions of the genome [47]. Accordingly, the Alu S subfamilies targeted by our assay are mainly found to occur in gene introns (Appendix A). However, as summarized in a previous review on the argument [3], the Alu elements which show a more variable, less stable methylation level are the relatively rare elements located in (promoter/intergenic) gene regulatory regions. In the Alu S subfamilies targeted by our assay, approximately five per cent of the elements overlap with transcription factor binding sites (in promoters or intergenic regions), hypothetically associated with gene regulation. Further studies, conducted using locus-specific methods for Alu methylation analysis, will be necessary to verify the specific contribute of the relatively rare Alu elements associated with gene-regulatory regions to the slight methylation differences that we observed between age groups.

Together with the known age-dependent expression changes of enzymes belonging to the DNA methylation/demethylation machinery [48,49], nutritional and dietary factors can affect Alu DNA methylation status in aging by changing the availability of the methyl donors and altering the activity of the DNMT enzymes [50,51]. 

Here, we found that reduced Alu CpG1 methylation levels were associated with lower consumption of whole-grain bread and fruit. Fruit is an important source of antioxidants and carotenoids [52], and our result, showing that a significant increase in fruit intake may improve Alu methylation status, is not surprising. In fact, several studies have found a link between global DNA hypomethylation and a low fruit consumption [53,54]. In addition, some evidence suggests that oxidative stress exposure increases Alu RNA expression in a retinal pigment epithelium cell lines (ARPE-19), while pre-treatment with carotenoids (lutein and zeaxanthin) significantly improve cell viability and reduce Alu RNA level, implying that these carotenoids have an effect on the Alu methylation [55]. In our analysis, we did not find any significant association between circulating levels of carotenoids and CpG1 Alu methylation, but this may be due to the relatively low sample size or to the impact of higher fruit consumption. With regard to the association between whole-grain bread consumption and Alu methylation there are no previous studies supporting this result. However, whole grains contain fibers, as well as a range of micronutrients and bioactive compounds, ref. [56] and whole grain wheat consumption may enhance antioxidant defense systems and reduce the risk of cancer [57].

Another finding of this study is the positive association between Alu CpG1 methylation and LDL2 cholesterol. This finding is consistent with previous reports regarding the role of lipid metabolism in DNA methylation. In fact, it has been demonstrated that a VLDL-and LDL-rich lipoprotein mix induces *de novo* DNA methylation in THP-1 macrophages via the activation of DNMT1 [58]. Other evidence reports that a high-fat diet promotes hypermethylation and decreases the transcription of hepatic genes controlling lipid homeostasis in a mouse model [59]. It also affects global DNA methylation in rat PBMCs and mouse heart [60,61]. Among metals only plasma Cu is positively associated with Alu CpG1 methylation. This finding is rather surprising as high circulating levels of Cu are associated with cardiovascular diseases [62] and cancer [63]. However, our population is exclusively represented by healthy subjects (both RASIG and GO) showing blood copper levels which are within the normal range (data not shown). Circulating Cu levels have been shown to improve macrophage antimicrobial function and on a physiological concentrations an increase of Cu levels can even be beneficial in the aging process [64]. It is notable that females display higher plasma Cu levels than males [65], and this fact may be related to their longer lifespan. 

The association of the increased methylation levels of Alu CpG2-5 with meat consumption should also deserve particular attention because of the reported health warnings regarding excessive meat consumption [66]. Moreover, these methylation sites display an opposite direction of the association with age compared to CpG1 (Appendix A). This data reinforces the idea of the existence of distinct epigenetic mechanisms operating at these different Alu methylation sites. 

## 5. Conclusions

In summary, our data support the hypothesis Alu methylation may be finely regulated by different mechanisms acting on different CpGs and that offspring of healthy nonagenarians may preserve Alu methylation status in particular at the CpG1 site. This may contribute to a better genome stability, thereby delaying the onset of age-related diseases. On the other hand, it has also been demonstrated that Alu RNA accumulation may exert a pro-atherogenic function by promoting oxidative stress and up-regulating pro-inflammatory mediators [67]. Based on our findings the genetic background of individuals from long-living families could have a major role in the maintenance of Alu methylation as opposed to dietary factors.

Furthermore, Alu CpG1 methylation may be affected by an increased consumption of fruit and whole-grain bread, or circulating LDL subfractions thus suggesting that targeted dietary interventions may be helpful in improving genomic stability and promoting longevity. Further research in a larger cohort and “in vitro” studies should be performed to support the present findings.

## Figures and Tables

**Figure 1 nutrients-11-02986-f001:**
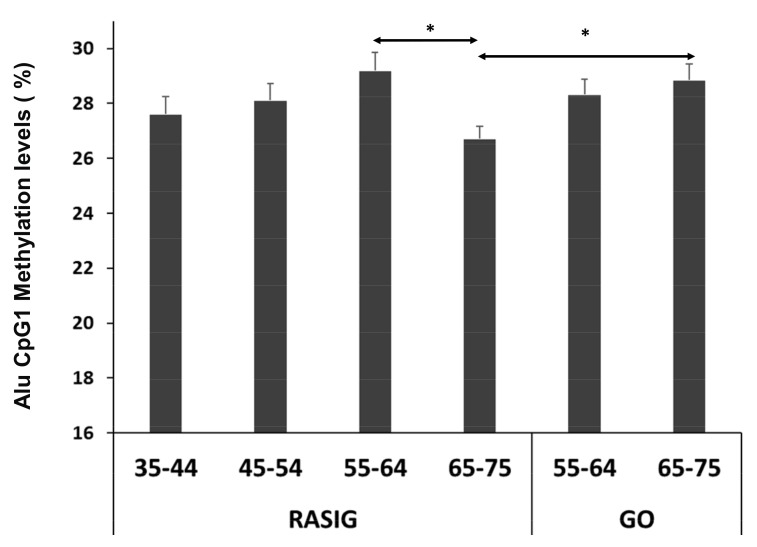
Alu CpG1 methylation in DNA extracted from whole blood of RASIG (n. 60) and GO donors (n. 32) recruited in Italy. RASIG showed lower Alu CpG1 methylation in the age class 65–75 years compared to RASIG in the age class 55-64 and to GO donors over 65 years. * *p* < 0.05.

**Figure 2 nutrients-11-02986-f002:**
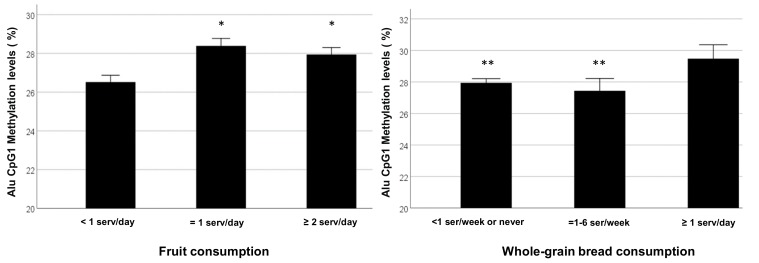
Influence of fruit and whole-grain bread consumption on Alu CpG1 methylation. A low consumption of fruit and whole-grain bread is associated with a reduced Alu CpG1 methylation level. * *p* < 0.05 as compared to fruit consumption<1 serv/day. ** *p* < 0.05 as compared to whole-grain bread consumption ≥1 serv/day.

**Table 1 nutrients-11-02986-t001:** Characteristics of subjects selected from the MARK-AGE project for Alu methylation study.

	RASIG (*n* = 60)					GO (*n* = 32)				
Age Classes (years)	35–44 (*n* = 12)	45–54 (*n* = 14)	55–64 (*n* = 11)	65–75 (*n* = 23)	*p* Value ^a^	55–64 (*n* = 17)	*p* Value ^b^	65–75 (*n* = 15)	*p* Value ^b^	*p* Value ^c^
Females (%)	4 (33.3%)	9 (64.3%)	6 (54.5%)	11 (47.8%)	NS	6 (35.3%)	NS	6 (40.0%)	NS	
RBC (×10^6^/μL)	5.01 ± 0.12	4.87 ± 0.08	4.77 ± 0.11	4.90 ± 0.10	NS	4.96 ± 0.10	NS	4.77 ± 0.11	NS	NS
Hemoglobine (g/dl)	14.30 ± 0.38	13.64 ± 0.41	14.51 ± 0.28	14.49 ± 0.30	NS	14.74 ± 0.34	NS	14.12 ± 0.30	NS	NS
WBC (×10^3^/μL)	6.16 ± 0.38	6.40 ± 0.48	6.08 ± 0.33	5.83 ± 0.27	NS	6.28 ± 0.44	NS	5.78 ± 0.25	NS	NS
Neutrophils (×10^3^/μL)	3.51 ± 0.26	3.61 ± 0.38	3.45 ± 0.29	3.46 ± 0.25	NS	3.79 ± 0.31	NS	3.28 ± 0.18	NS	NS
Lymphocytes (×10^3^/μL)	1929 ± 148	2008 ± 118	1892 ± 186	1764 ± 96	NS	1790 ± 151	NS	1776 ± 90	NS	NS
Monocytes (×10^3^/μL)	364 ± 38	401 ± 30	371 ± 30	366 ± 19	NS	357 ± 31	NS	334 ± 24	NS	NS
Platelets (×10^3^/μL)	241 ± 50	289 ± 67	278 ± 74	226 ± 70	0.031	238 ± 44	NS	248 ± 52	NS	NS
CRP (μg/L)	1.23 ± 0.39	1.13 ± 0.32	1.25 ± 0.40	1.87 ± 0.35	NS	1.65 ± 0.24	NS	3.18 ± 0.97	NS	NS
TC (mmol/L)	5.02 ± 0.23	5.22 ± 0.23	6.28 ± 0.31	5.75 ± 0.30	0.045	5.54 ± 0.18	0.041	5.71 ± 0.19	NS	NS
HDL (mmol/L)	1.31 ± 0.10	1.38 ± 0.12	1.51 ± 0.08	1.49 ± 0.12	NS	1.37 ± 0.10	NS	1.47 ± 0.11	NS	NS
LDL (mmol/L)	2.92 ± 0.16	3.18 ± 0.23	4.07 ± 0.22	3.47 ± 0.24	0.035	3.44 ± 0.15	0.030	3.52 ± 0.18	NS	NS
TG (mmol/L)	2.20 ± 0.90	1.47 ± 0.40	1.30 ± 0.13	1.49 ± 0.20	NS	1.14 ± 0.53	NS	1.15 ± 0.46	NS	NS
FG (mmol/L)	5.15 ± 0.29	5.30 ± 0.16	5.43 ± 0.11	5.82 ± 0.16	NS	5.82 ± 0.34	NS	5.88 ± 0.31	NS	NS
Creatinine (μmol/L)	73.2 ± 5.4	65.7 ± 4.1	72.3 ± 5.0	75.4 ± 3.1	NS	73.3 ± 3.8	NS	69.9 ± 3.0	NS	NS
BMI	25.7 ± 1.3	24.8 ± 1.4	26.3 ± 1.1	28.9 ± 0.9	NS	25.7 ± 1.6	NS	27.1 ± 0.9	NS	NS
Smoking Never	7 (58.3%)	7 (50.0%)	4 (36.4%)	14 (60.9%)	NS	9 (52.9%)	NS	9 (60.0%)	NS	NS
Former	2 (16.7%)	5 (35.7%)	5 (45.5%)	8 (34.8%)		6 (35.3%)		3 (20.0%)		
current	3 (25.0%)	2 (14.3%)	2 (18.2%)	1 (4.3%)		2 (11.8%)		3 (20.0%)		
Alchol consumption < 1 serv./day	10 (83.3%)	10 (71.4%)	5 (45.5%)	17 (73.9%)	NS	10 (58.8%)	NS	7(46.7%)	NS	NS
=1 serv./day	1 (8.3%)	2 (14.3%)	2 (18.2%)	1 (4.3%)		1 (5.9%)		4 (26.7%)		
>1 serv./day	1 (8.3%)	2 (14.3%)	4 (36.4%)	5 (21.7%)		6 (35.3%)		4 (26.7%)		

Data are reported as mean ± Standard Error of the Mean (SEM) for continuous variables or N (%) for categorical variables. RBC Red Blood Cells, WBC white blood cells, CRP C-reactive protein, TG triglycerides, TC total cholesterol, HDL high-density lipoprotein cholesterol; LDL Low- density lipoprotein cholesterol, FG fasting glucose, BMI Body mass index. ^a^
*p*-value from ANOVA (Bonferroni as post-hoc test) or Kruskas Wallis test (Dunn test as post hoc analyses) (continuous variables) and Chi-square test (categorical variables) comparing RASIG age-classes. ^b^
*p*-value from ANOVA or Kruskas Wallis test (continuous variables) or Chi-square test (categorical variables) compared to RASIG in the same age class. ^c^
*p*-value from ANOVA or Kruskas Wallis test (continuous variables) and Chi-square test (categorical variables) comparing GO age-classes.

**Table 2 nutrients-11-02986-t002:** Automatic linear regression analysis for variables independently associated with Alu CpG1 methylation in RASIG and GO donors.

Predictors	Coefficient	Std. Error	Importance	Sig
Subject group ^a^	2.219	0.542	0.200	0.0001
Plasma Cu	0.0005	0.001	0.156	0.001
LDL2-C ^b^	0.043	0.015	0.104	0.004
Ascorbic acid	−0.218	0.080	0.088	0.008
Total Glutathione	−0.002	0.001	0.073	0.015
Whole-grain bread consumption ^c^	−1.906	0.804	0.067	0.020
Age	−0.054	0.024	0.063	0.024
HDL2-C	0.081	0.037	0.057	0.032
Plasma Zn	−0.005	0.002	0.054	0.036
Fruit consumption ^c^	−1.687	0.806	0.052	0.040

^a^ Rasig group represents the reference. ^b^ LDL2-C, indicates cholesterol in LDL2 subfraction; HLDL2-C indicates cholesterol in HDL2 subfraction. ^c^ For fruit consumption: =1 serv/day and ≥2 serv/day were automatically combined, used as reference and compared to <1 serv/day; For whole-grain bread consumption: 1–6 serv/week and ≥1 serv/day were automatically combined, used as reference and compared to <1 serv/week.

**Table 3 nutrients-11-02986-t003:** Generalized linear regression model for variables independently associated with Alu CpG1 methylation in RASIG and GO donors.

Predictors	Coefficient	Std. Error	Sig
Subject group ^a^	0.070	0.022	0.002
Plasma Cu	0.153	0.052	0.005
LDL2-C ^b^	0.078	0.039	0.048
Whole-grain bread consumption (1-6 serv/week) ^c^	−0.086	0.037	0.024
Whole-grain bread consumption (≥ 1 serv/day) ^c^	−0.118	0.042	0.007
Fruit consumption (≥ 1 serv/day) ^d^	−0.070	0.035	0.046

^a^ RASIG group represents the reference. ^b^ LDL2-C, indicates cholesterol in LDL2 subfraction. ^c^ consumption <1 serv/day was taken as the reference. ^d^ consumption < 1 serv/day was taken as the reference.

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
