# Peer review of "Nutritional Factors Modulating Alu Methylation in an Italian Sample from The Mark-Age Study Including Offspring of Healthy Nonagenarians"

_nutrients, 2019, doi:10.3390/nu11122986_

Round 1
Reviewer 1 Report
The manuscript by Giacconi et al is well written and clear. Additionally, the sample cohort itself and the nutritional information from the participants are of high interest. However, I feel that the DNA methylation data they present is very limited to support their conclusions and I suggest the authors to include more methylation markers in the study. Also, I miss relevant information in the Methods section.
Specific comments:
The authors mention PBMCs in the first section of Methods but it seems that they recruit and isolate them but they do not use them at any stage in this study. Please clarify. Sections 2, 3, 4 and 5 in Methods lack detail. I would include at least a brief explanation in each section. I think that the authors should provide more information about the Alu sequence they are analyzing in Methods. How many copies do you estimate to be analyzing across the genome? Or is it a unique sequence? If so, where is it located? The authors provide more details in Discussion but I think it is important to introduce this before. In Methods, the authors mention that they are analyzing only the first five positions. Why? Is it due to a technical reason or is there any other reason? Please specify. The authors mention several controls: an oligonucleotide from Qiagen (what for?) and fully methylated/unmethylated controls but do not mention the results. Did the authors consider using a calibration curve? Results are clearly presented but methylation data seems very weak and limited to draw any conclusions. In Discussion, the authors mention that they found a significant trend towards an increased methylation with age in one of the CpGs. Considering the weakness of the results for CpG1, I think that this increased methylation could be as real as the results obtained for CpG1, which have been followed up with much more interest. I think we should not focus only on those results that go in line with our aprioristic ideas… In Discussion, again, I miss the genomic location of the Alu sequence. My main comment is that the cohort and the collected information are very interesting but that the methylation data is too weak and limited to support any conclusions. I suggest that the authors include additional markers or even explore the whole methylome (the authors could start considering for example the CpG positions of the Epigenetic-Horvard or the Mitotic-Teschendorff clocks).Author Response
The authors mention PBMCs in the first section of Methods but it seems that they recruit and isolate them but they do not use them at any stage in this study. Please clarify.
We apologize for this mistake; we had erroneously reported the PBMC procedure from MARK-AGE project while in the present paper the material used to assess Alu methylation was whole blood (and plasma or serum for the additional nutritional biomarkers). We have now corrected this mistake.
Sections 2, 3, 4 and 5 in Methods lack detail. I would include at least a brief explanation in each section. I think that the authors should provide more information about the Alu sequence they are analyzing in Methods. How many copies do you estimate to be analyzing across the genome? Or is it a unique sequence? If so, where is it located? The authors provide more details in Discussion but I think it is important to introduce this before. In Methods, the authors mention that they are analyzing only the first five positions. Why? Is it due to a technical reason or is there any other reason? Please specify.
As suggested by the reviewer, we have now implemented the Section of methods, reporting the detailed methodologies in red from page 6 to 9 of the manuscript. However, for the sections regarding the determination of total glutathione, total free cysteine, ascorbic acid and uric acid and total carotenoids the methods were already described in detail in the following publications (Weber, D. et al, Oxidative Medicine and Cellular Longevity 2017, 2017, 1–12; Stuetz, W.; et al., Nutrients 2016, 8, 614). In those publications, the same samples from the MARK-AGE Project, as in this manuscript, were used and also the analyses were exactly the same, carried out on the same HPLC system by Stuetz team. Therefore, we believe it is redundant to describe the methods in the same detail as previously.”
Concerning the Alu elements targeted by the used bisulfite pyrosequencing method, the number of single genomic elements contributing to the CpG methylation values obtained in our analysis may be expected to be in the order of ten thousand to hundreds of thousands. In order to follow the suggestion of the reviewer, we greatly extended, in the Methods section and in supplemental material, the description of Alu elements expected to be targeted by the assay. In particular, we added: (1) a detailed description of the Alu subfamilies which, based on their consensus sequences, are predicted to be targeted by the used primers; (2) specific data about the genomic localization of the elements of these subfamilies, with respect to genes and regulatory regions; (3) a reference to a previous work (Yang et al., 2004) in which the actual number of Alu elements amplified by a similar bisulfite PCR approach has been estimated.
The authors mention several controls: an oligonucleotide from Qiagen (what for?) and fully methylated/unmethylated controls but do not mention the results. Did the authors consider using a calibration curve?
Following the Reviewer’s suggestion, we specified the use we made of the control oligonucleotide, and we added more complete information about the pyrosequencing method in the supplemental data section. In addition, we now added in supplemental data the results of the calibration curves conducted by using different proportions of fully methylated and fully unmethylated control DNA (Supplementary Figure 3S).
Results are clearly presented but methylation data seems very weak and limited to draw any conclusions.
Indeed, we claim that this is a preliminary study (addressed to the specific thematic of the Special Issue). However, these preliminary findings are important to drive further studies, not only for the proposed nutritional factors that are able to modulate Alu methylation but also because of the differential regulation observed by different methylation sites of Alu (for which the number of subjects is enough, as highly correlated methylation sites are easily detected in lower number of subjects).
In Discussion, the authors mention that they found a significant trend towards an increased methylation with age in one of the CpGs. Considering the weakness of the results for CpG1, I think that this increased methylation could be as real as the results obtained for CpG1, which have been followed up with much more interest. I think we should not focus only on those results that go in line with our aprioristic ideas…
We deem this is a proper observation. In the revised version we have given more attention to the increase of CpG2-5 with age and its association with nutrients (Supplementary table 6S) and have discussed the possible rationale behind this phenomenon. However, we decided to focus the majority of the results on CpG1 as this methylation marker was the sole displaying a significant difference between RASIG and GO at comparable age.
In Discussion, again, I miss the genomic location of the Alu sequence. My main comment is that the cohort and the collected information are very interesting but that the methylation data is too weak and limited to support any conclusions. I suggest that the authors include additional markers or even explore the whole methylome (the authors could start considering for example the CpG positions of the Epigenetic-Horvard or the Mitotic-Teschendorff clocks).
Following the suggestion of the Reviewer, in the new version of the manuscript we added, in the Methods and in the Discussion, considerations about the genomic location of the Alu elements. We also considered the possible relationship between the genomic location of Alu elements and the stability (or variability) of their methylation status.
We agree that the conclusions of this manuscript regarding the biological age of the subjects are weak and would have required the use (for example) of an epigenetic clock. So, we revised the sentences related to these conclusions. However, the focus of the manuscript was to investigate nutritional factors (from a wide panel) that are mostly related to Alu methylation in order to open the way to further studies addressing these nutrients. In the revised version we have additionally reported nutritional factors affecting the other CpG sites. Importantly, we also noticed a difference in the direction of the association with age of CpG1 (Table 2) versus CpG2-5 (Supplementary Table 6S). This data supports the idea that these methylation sites are differently regulated and, at our knowledge, this is the first time that this phenomenon is observed. Likely, we were able to observe this phenomenon due to the high-resolution method (Pyrosequencing assay) used in this study, whereas classic methylome approach, such as the bead array, exclude highly repetitive regions such as Alu CpG sites.
Reviewer 2 Report
-Line 68: What it means “appear biologically younger”. The authors should clarify this sentence.
-Lines 87-92: please add the ethical approval that must list the authority that provided approval and the corresponding ethical approval code.
-Improve the quality of Figures 1 and 2
-Line 194 : there is typing mistake
-I don’t understand the relevance of 35-44 and 45-54 RASIG groups. There are not equivalent groups for GO cohort. For me, it will be more relevant to use only 55-64 and 65-75 groups. In addition, there is no significant difference between 65-75 RASIG group and 35-44 and 45-54 RASIG groups for ALU GpG1methylation (figure 1). Then, it seems to be that there is no decrease of ALU GpG1 methylation with age.
- This study is interesting but I am very puzzled by the size of the sample. Is it possible for the authors to include all the subjects or other subjects of the MARK-AGE project ?
Author Response
Line 68: What it means “appear biologically younger”. The authors should clarify this sentence.We have now clarified the concept reporting the following sentence: “centenarians’ offspring show decreased epigenetic age of PBMCs than those of age matched controls”
2) Lines 87-92: please add the ethical approval that must list the authority that provided approval and the corresponding ethical approval code.
At page 6 of the method section we have reported the following sentences “Local Ethics Committees approval for the recruitment of Italian participants in MARK-AGE project was obtained by S.Orsola-Malipighi Hospital Committee (approved the project on 24/06/2008 code n. 75/2008/U/Tess) and AUSL – Independent Ethical Commettee (Emilia-Romagna Region) approved on 01/04/2009– Prot num 346/CE; CE: 09007.
3) Improve the quality of Figures 1 and 2
We have now improved the quality of Figures 1 and 2 and uploaded as separate files.
4) Line 194 : there is typing mistake
The typing mistakes along the manuscript have been corrected.
5) I don’t understand the relevance of 35-44 and 45-54 RASIG groups. There are not equivalent groups for GO cohort. For me, it will be more relevant to use only 55-64 and 65-75 groups. In addition, there is no significant difference between 65-75 RASIG group and 35-44 and 45-54 RASIG groups for ALU GpG1methylation (figure 1). Then, it seems to be that there is no decrease of ALU GpG1 methylation with age.
We agree with this observation and have changed the interpretation of the results and the relative discussion. In particular, we have now specified that CpG1 levels do not display large changes with age but are lower in the oldest subjects of the RASIG population (65-75 years) compared to RASIG (55-64 years). Conversely, GO do not display the same change and their levels at 65-75 years are more similar to those of the 55-64 years’ RASIG. However, we should also take into account the results of Table 2 and the new Supplementary Table 6S which suggest an opposite association with age for the two methylation markers. Being the results from an automatic linear regression model it is also relevant that only CpG1, and not the other methylation markers, is found to be related to subject group (GO or RASIG).
This study is interesting but I am very puzzled by the size of the sample. Is it possible for the authors to include all the subjects or other subjects of the MARK-AGE project ?We indeed plan to perform further studies in larger populations in the future. We would have no chance to include additional subjects with the time restrains given to us by the editors for this revision (10 days). However, we deem that the findings of this preliminary study may be important to address the future focus towards epigenetic regulation of Alu at different CpG sites and towards specific nutritional factors that may interfere with Alu epigenetics and with potential interest for health. As also suggested by reviewer 1 we also added al lot of information regarding the Alu methylation assay (Figure 2S, Figure 3S, Figure 4S, Table 1S, Table 2S) that might be useful to develop locus-specific methods for Alu methylation analysis in future studies.

Round 2
Reviewer 1 Report
The manuscript has considerably improved, however, my main concern keeps being the same. I consider that analyzing the methylation levels of a unique locus is a too preliminary study to be published, and I do not think many conclusions can be drawn from a unique experiment. I think that a better characterization of the methylation levels of Alu sequences (and not only 5 CpGs) is the minimum to be done so that the paper can be considered for publication.
Reviewer 2 Report
Despite a size sample that could be better, Giacconi et al. have addressed my concerns. I have nofurther questions or concerns to add and I feel the manuscript is suitable with the scope of the journal.